# Epithelioid Cutaneous Mesenchymal Neoplasms: A Practical Diagnostic Approach

**DOI:** 10.3390/diagnostics10040233

**Published:** 2020-04-17

**Authors:** Joon Hyuk Choi, Jae Y. Ro

**Affiliations:** 1Department of Pathology, Yeungnam University College of Medicine, Daegu 42415, Korea; 2Department of Pathology and Genomic Medicine, Houston Methodist Hospital, Weill Medical College of Cornell University, Houston, TX 77030, USA; JaeRo@houstonmethodist.org

**Keywords:** skin neoplasms, diagnosis, epithelioid cells, sarcoma, melanoma, carcinoma

## Abstract

Epithelioid cells are rounded or polygonal cells with abundant eosinophilic or clear cytoplasm and ovoid to round nuclei, superficially resembling epithelial cells. Cutaneous mesenchymal neoplasms composed predominantly or exclusively of epithelioid cells are relatively uncommon and can cause considerable diagnostic difficulties due to overlapping histologic features among heterogeneous groups of tumors. Familiarity with practical diagnostic approaches and recognition of key histopathologic features are important for correct diagnosis and management. This review summarizes the histologic features of epithelioid cutaneous mesenchymal neoplasms and discusses their differential diagnoses from malignant melanomas and carcinomas.

## 1. Introduction

Epithelioid mesenchymal neoplasms are predominantly or extensively comprised of rounded or polygonal tumor cells resembling epithelial cells [1]. Epithelioid mesenchymal neoplasms of the skin include a heterogeneous group of tumors with divergent lineages. The incidence of epithelioid cutaneous mesenchymal neoplasms is low, and they are often challenging to diagnose because of their morphological overlap with various tumor types. Therefore, careful morphologic evaluation is necessary to guide proper diagnosis, patient management. A well-organized diagnostic approach can help pathologists achieve specific differential diagnoses or histologic mimics. Here we review the histologic features of epithelioid cutaneous mesenchymal neoplasms and offer a practical approach to their differential diagnosis.

## 2. Practical Diagnostic Approach to Epithelioid Cutaneous Mesenchymal Neoplasms

### 2.1. Skin Anatomy

The skin is a large, heterogeneous organ that accounts for about 15% of total body weight, making the largest body organ [2]. The skin is composed of three layers: (1) Epidermis, (2) dermis, and (3) the subcutaneous adipose tissue. Each of these layers has a unique function and structure that depends on age, sex, race, and anatomic location. Basic knowledge of normal cutaneous histology is essential for an understanding of its pathologic states.

Anatomic sites are an important factor in the differential diagnoses of potential cutaneous mesenchymal neoplasms as certain tumors occur more frequently at particular sites. For example, cutaneous angiosarcomas most commonly occur in sun-damaged skin of the head and neck in elderly patients [3], and sarcomas frequently arise in the therapeutic irradiation fields.

### 2.2. Clinical Considerations

Cutaneous benign and malignant soft tissue tumors usually present as painless masses exhibiting variable growth rates [4]. Cutaneous lesions with plaques or elevated nodules can be ulcerated when they are malignant. The majority of benign mesenchymal tumors are superficial and <5 cm in size. Any patient presenting with a soft tissue mass that is increasing in size or >5 cm should either be referred for an urgent ultrasound scan or directly to a sarcoma diagnostic center, regardless of whether the mass is painful [5]. Radiological findings can be helpful to determine the involvements of subfascial soft tissue or skeletal muscle.

Pathologists should be aware of relevant clinical information including patient age, clinical symptoms, and the duration, size, and location of their tumor. Sarcomas constitute only 1% of major malignancies in adults [6]. Common benign soft tissue tumors that arise in the skin are dermatofibromas, lipomas, vascular or smooth muscle tumors, and nerve sheath tumors [4]. Cutaneous angiosarcoma, myxofibrosarcoma, undifferentiated pleomorphic sarcoma, and leiomyosarcoma arise predominantly in elderly patients, whereas dermatofibrosarcoma protuberans, epithelioid sarcoma, and clear cells sarcoma are more common in younger patients [7].

### 2.3. Histologic Evaluation

Careful microscopic examination of hematoxylin and eosin (H&E)-stained tumor sections should be performed at low-power magnification to evaluate the lesion’s: (1) Size and depth, (2) relation to overlying skin, (3) border characteristics (e.g., pushing, infiltrative), (4) architectural (growth) pattern, and (5) stromal features. Tumor cell morphology, tumor necrosis, and mitotic activity should be evaluated at high-power magnification. Malignant soft tissue tumors are generally characterized by nuclear atypia with pleomorphism, mitotic activity, and necrosis.

In terms of practical diagnostic resources, a pattern-based approach is a useful technique that substantially aids the diagnostic process [8,9]. Epithelioid cutaneous mesenchymal neoplasms show variable growth patterns, that included lobulated, nested, trabecular, cord-like, and sheet-like architectures (Table 1). In addition, some tumors have characteristic cytoplasmic features (e.g., clear or granular) and stromal features (e.g., prominent inflammatory cell infiltration). Thus, an understanding of architectural patterns, cytoplasmic quality, and stromal features can facilitate proper diagnosis.

### 2.4. Immunohistochemistry

Soft tissue tumors are traditionally classified according to the line of differentiation. Immunohistochemistry plays a critical role in the identification of the line of differentiation and diagnosis of soft tissue tumors that lack distinct morphologic features. The recommended immunohistochemical panel for epithelioid cutaneous mesenchymal neoplasms includes CD34, smooth muscle actin (SMA), S100 protein, ALK, SMARCB1 (Switch/sucrose non-fermentable (SWI/SNF)-related, matrix associated, actin-dependent regulator of chromatin, subfamily B, member 1 also called integrase interactor 1 (INI1)), vascular markers (e.g., CD31, ERG, FLI1), melanocytic markers (e.g., HMB-45, melan-A), and epithelial markers (e.g., cytokeratin (AE1/AE3), EMA) (Table 2).

Though many immunohistochemical markers traditionally have limited specificities, more specific immunohistochemical markers have recently been developed for benign and malignant mesenchymal neoplasms [10,11]. However, pathologists should continue to interpret immunohistochemistry results carefully in context.

### 2.5. Molecular Diagnostic Advancements

Recent advances in the classification of cutaneous mesenchymal neoplasms include description of recurrent chromosomal and genetic alterations in a significant subset of soft tissue tumors [12]. The SWI/SNF complex is involved in chromatin remodeling and transcriptional regulation, and SMARCB1 (INI1) is a member of the SWI/SNF complex family. Inactivating mutations in the *SMARCB1* (*INI1*) and *SMARCB4* genes were recently described in a subset of soft tissue tumors with epithelioid or rhabdoid cell features [13,14,15]. Molecular diagnostic studies continue to enrich our understanding of the biological mechanisms of tumor development and progression. Molecular testing plays a critical role in the diagnosis of benign and malignant soft tissue tumors and should be implemented routinely for soft tissue tumor diagnosis. Specific genetic alterations identified in epithelioid cutaneous mesenchymal neoplasms are summarized in Table 3.

## 3. Epithelioid Fibroblastic/Myofibroblastic and So-Called Fibrohistiocytic Tumors

### 3.1. Epithelioid Fibrous Histiocytoma

Epithelioid fibrous histiocytoma (EFH) is a distinctive, usually exophytic, well circumscribed benign cutaneous neoplasm composed of epithelioid cells (including binucleated forms) [16]. The vast majority of cases show *ALK* rearrangement [17,18,19]. EFH is considered a biologically distinct tumor type that is unrelated to conventional fibrous histiocytoma, and usually arises in young and middle-aged adults (mean patient age: 40 years). EFH predominantly affects the lower and upper extremities, followed by the trunk, head, and neck, and usually presents as a single exophytic nodule, often with a vascular appearance.

Histologically, tumors are composed of uniform plump epithelioid cells with round to oval, vesicular nuclei, small nucleoli, and abundant eosinophilic cytoplasm, and occasionally binucleated cells. Often tumors are surrounded by an epidermal collarette. Prominent small thin-walled blood vessels may be present. Recently, rare variants composed of spindled, non-epithelioid cells have been reported [20]. Immunohistochemically, ALK is overexpressed in approximately 90% of cases [17] and EMA is positive in approximately 60% of cases [21].

The differential diagnoses for EFH include cutaneous syncytial myoepithelioma, epithelioid sarcoma, and spitzoid melanocytic lesions. Cutaneous syncytial myoepithelioma is positive for S100 protein and negative for ALK, whereas epithelioid sarcoma shows greater nuclear atypia than EFH and loss of SMARCB1 (INI1) expression. Spitzoid melanocytic lesions such as Spitz nevi and spitzoid melanomas show a junctional component and are positive for S100 protein and melanocytic markers.

### 3.2. Myxoinflammatory Fibroblastic Sarcoma

Myxoinflammatory fibroblastic sarcoma (MIFS) is a low-grade fibroblastic neoplasm characterized by prominent inflammation, bizarre ganglion-like cells, and vacuolated, mucin-filled cells (pseudolipoblasts) [22]. MIFS harbors t(1;10)(p22;q24) with *TGFRB3-MGEA5* gene fusion and exhibits amplification of chromosome 3p11-12, which is associated with elevated expressions of *VGLL3* and *CHMP2B* [23,24,25]. MIFS occurs in middle-aged adults and shows no sex predilection. In most cases, it occurs in distal extremities (fingers, hands, and feet), and usually presents as a slow-growing, painless mass resembling a benign condition [26]. The tumor is poorly demarcated, recurrences are common, and metastases are rare [26]. Complete surgical excision provides the best chance of disease-free survival.

Histologically, the tumor is composed of myxohyaline and collagenous zones, with a multinodular appearance. The variable, atypical, spindle to epithelioid tumor cells are admixed with prominent inflammatory cell infiltrate (Figure 1). Occasional pleomorphic, mono- or multinucleated large cells with prominent viral inclusion-like nucleoli (Reed-Sternberg-like cells) or ganglion-like cells are present. Immunohistochemically, the tumor cells show variable positivity for D2-40, CD68, CD34, and SMA [27].

The differential diagnoses of MIFS include inflammatory or infectious lesions, hemosiderotic fibrolipomatous tumor (HFLT), myxofibrosarcoma, and superficial CD34+ fibroblastic tumor. However, inflammatory and infectious lesions lack bizarre nuclear atypia, and HFLT lacks Reed-Sternberg-like cells and prominent inflammation. Recently, it was suggested that HFLT is closely related to MIFS [28]. Myxofibrosarcoma shows distinctive curvilinear vascular patterns and less conspicuous inflammatory infiltrates. Superficial CD34+ fibroblastic tumors are more cellular than MIFS and diffusely positive for CD34 [29].

## 4. Epithelioid Smooth Muscle, Perivascular, and Vascular Tumors

### 4.1. Cutaneous Smooth Muscle Tumors

Cutaneous leiomyosarcoma (atypical intradermal smooth muscle tumor) is a dermal mesenchymal neoplasm of intermediate malignancy and smooth muscle differentiation [30]. It often occurs in the trunk and extremities of old adults. Local recurrence is common, usually due to incomplete excision. However, when the tumors are confined to the dermis, they do not metastasize [31,32]. Clinically, it present as a small, solitary, and occasionally painful nodule.

Histologically, cutaneous leiomyosarcoma shows atypical spindled cells arranged in fascicular pattern. The tumor cells have elongated, blunt-ended nuclei with brightly eosinophilic cytoplasm. Nuclear enlargement, hyperchromatism, and mitotic activity are present. Epithelioid leiomyosarcomas of the skin have been rarely described [33]. Immunohistochemically, the tumor cells are positive for SMA, desmin, and h-caldesmon.

The differential diagnoses of epithelioid leiomyosarcoma include leiomyoma, epithelioid rhabdomyosarcoma, epithelioid sarcoma, malignant melanoma, and carcinomas. Cutaneous leiomyomas lack nuclear atypia and mitoses. Epithelioid rhabdomyosarcoma is a recently described variant of rhabdomyosarcoma [34]. Primary cutaneous epithelioid rhabdomyosarcomas have been described [35]. Epithelioid rhabdomyosarcomas are positive for specific markers of skeletal muscle differentiation (e.g., myogenin, MyoD1). Epithelioid sarcomas are positive for CD34 and cytokeratin and show loss of INI1 expression. Malignant melanomas are positive for S100 protein and melanocytic markers. Primary and metastatic carcinomas are positive for epithelial markers (e.g., cytokeratin, EMA).

### 4.2. Glomus Tumor

Glomus tumors are mesenchymal neoplasms composed of cells resembling modified smooth muscle cells of the normal glomus body [36]. Glomus tumors of skin are more common than their soft tissue counterparts, and most frequently occur in the distal extremities, including the subungual region, digits, hand, wrist, or foot [37]. These tumors commonly arise in the fourth to sixth decades of life. Most cutaneous glomus tumors present as solitary, small (<1 cm), painful nodules or plaques, with a pink to blue hue. Multiple familial glomus tumors show an autosomal dominant pattern [38].

Histologically, the tumor cells are small, uniform, and round with a central round nucleus, amphophilic to lightly eosinophilic cytoplasm, and a sharp cellular outline with the peripheral basal lamina. Epithelioid variants composed of large polygonal cells with abundant eosinophilic cytoplasm have been reported [39]. Malignant glomus tumors are extremely rare, and the diagnosis of malignant glomus tumors should be reserved only for tumors showing marked nuclear atypia and any level of mitotic activity, or atypical mitotic figures [40]. Cutaneous malignant glomus tumors follow a more indolent clinical course than their deep soft tissue counterparts [41]. Immunohistochemically, the tumor cells typically express SMA and pericellular collagen type IV. Transducing-like enhancer of split 1 (TLE1) is a potential immunohistochemical marker for glomus tumors [42]. *MIR143-NOTCH* gene fusion and *NOTCH* gene rearrangements have been reported in benign and malignant glomus tumors [43].

The differential diagnoses of glomus tumor include myopericytoma, benign adnexal tumors, and melanocytic nevus. Myopericytomas show characteristic perivascular, concentric growth of ovoid to spindle tumor cells. Myopericytomas represent a morphologic spectrum with myofibromas, angioleiomyomas, and glomus tumors. Benign adnexal tumors (e.g., nodular hidradenoma) show epithelial differentiation and are positive for cytokeratin and EMA, whereas melanocytic nevi show nests of melanocytic cells and are positive for S100 protein and melanocytic markers.

### 4.3. Epithelioid Hemangioma

Epithelioid hemangioma is a benign vascular proliferation that often has an associated inflammatory component. The vessels within the tumor are lined by prominent endothelial cell with epithelioid morphology [44]. These tumors are most common in young adults, arise over a wide age range, and show a slight male predilection [45]. Epithelioid hemangioma occurs in the head and neck and distal extremities (digits). *FOS* gene rearrangements are present in the majority of bone and soft tissue epithelioid hemangiomas but are rare in cutaneous epithelioid hemangiomas [45,46]. Lesions are single or multiple, persistent, skin-colored to erythematous papules and nodules.

In most cases, tumors are histologically well-demarcated and show lobular proliferation of vascular channels. The channels are lined by bland epithelioid endothelial cells with vesicular nuclei and abundant eosinophilic or amphophilic cytoplasm. These cells may project into the lumen and have a hobnail or tombstone-like appearance. Inflammatory cells, including eosinophils, lymphocytes, plasma cells, and histiocytes, infiltrate the stroma profusely. Exuberant or atypical examples of epithelioid hemangiomas have nest or sheet-like aggregates of epithelioid endothelial cells (Figure 2) [47]. Immunohistochemically, endothelial cells are positive for vascular markers and are accompanied by an SMA-positive pericyte layer. Cutaneous cellular epithelioid hemangiomas show diffuse strong FOS-B expression [48].

The differential diagnoses of epithelioid hemangioma include Kimura disease, epithelioid hemangioendothelioma, and epithelioid angiosarcoma. Kimura disease usually affects young Asian men and does not have vessels lined by epithelioid endothelial cells [49]. Epithelioid hemangioendotheliomas show nests or cords of tumor cells in a distinctive myxohyaline stroma, and epithelioid angiosarcomas show marked nuclear atypia and infiltrative growth.

### 4.4. Pseudomyogenic Hemangioendothelioma

Pseudomyogenic hemangioendothelioma is a spindled to epithelioid, rarely metastasizing endothelial neoplasm that often presents as multiple discontiguous nodules in different tissue planes [50]. The tumor shows t(7;19)(q22;q13) with *SERPINE1-FOSB* fusion [51]. Pseudomyogenic hemangioendothelioma usually affects young adults and shows a male predominance. It arises in the lower extremities, upper extremities, or trunk [52], and is multifocal in two-thirds of cases, often involving multiple tissue planes (e.g., skin, subcutis, skeletal muscle, and bone). Clinically, these lesions may be painless or painful nodules.

Histologically, the tumor has infiltrative borders and is composed of loose fascicles of mildly atypical plump spindled to epithelioid cells with brightly eosinophilic cytoplasm and, in some cases, a rhabdomyoblast-like appearance. Prominent stromal neutrophilic infiltrate is present in about 50% of cases [52]. Immunohistochemically, the tumor cells are positive for CD31, ERG, FLI1, and cytokeratin (AE1/AE3), but negative for CD34. The tumor cells also show diffuse nuclear immunoreactivity for FOSB [53,54].

The differential diagnoses of pseudomyogenic hemangioendothelioma include cellular dermatofibroma, epithelioid sarcoma, and spindle cell squamous cell carcinoma. Cellular dermatofibromas lack plump, myoid-appearing tumor cells and are negative for CD31 and cytokeratin. Epithelioid sarcomas often show nodular growth with central necrosis and loss of SMARCB1 (INI1) expression. Spindle cell squamous cell carcinomas have dysplasia or a carcinoma *in situ* component in the overlying epidermis and are negative for CD31.

### 4.5. Epithelioid Hemangioendothelioma

Epithelioid hemangioendothelioma is a borderline malignant endothelial neoplasm composed of epithelioid cells in myxohyaline stroma [50]. Cutaneous epithelioid hemangioendothelioma is rare and usually occurs in middle-aged adults [55,56]. It arises in a variety of sites, including the extremities, head, neck, and back. The majority of epithelioid hemangioendotheliomas harbor t(1;3)(p36;q25) with *WWTR1-CAMTA1* gene fusion [57], although a small subset has t(X;11)(p11;q22) with *YAP1-TFE3* gene fusion [58]. Clinically, the neoplasm usually presents as a slightly raised, erythematous, sometimes painful dermal nodule.

Histologically, the tumor is composed of relatively bland epithelioid cells arranged in cords, nests, or a trabecular pattern in variably myxohyaline stroma (Figure 3). Intracytoplasmic vacuoles are often seen. Immunohistochemically, the tumor cells are positive for vascular markers (e.g., CD31, CD34, FLI1, and ERG). Cytokeratins are expressed in 20%–30% of cases. CAMT1 is a useful immunohistochemical marker for distinguishing epithelioid hemangioendothelioma from histologic mimics [59].

The differential diagnoses of epithelioid hemangioendothelioma include epithelioid angiosarcoma, epithelioid sarcoma, and metastatic carcinomas. Epithelioid angiosarcomas show more cytologic atypia than epithelioid hemangioendotheliomas and are negative for CAMTA1. Epithelioid sarcomas lack intracytoplasmic vacuoles and do not express SMARCB1 (INI1). Metastatic carcinomas of the skin show a higher degree of cytologic atypia with necrosis and are negative for vascular markers.

## 5. Epithelioid Nerve Sheath Tumors

### 5.1. Epithelioid Schwannoma

Schwannoma is a benign neoplasm composed of Schwann cells [60]. Epithelioid schwannoma is a rare variant of schwannoma and is composed predominantly or exclusively of epithelial-appearing Schwann cells. This neoplasm usually occurs in adults [61], and principally in the dermis/subcutis of the upper and lower extremities, thorax, abdomen, and back [62]. Malignant changes in epithelioid schwannoma have been reported [63]. Clinically, it presents as a painless, palpable mass.

Histologically, the tumor is well-circumscribed, encapsulated, and composed of epithelioid or polygonal Schwann cells arranged in sheets and nests (Figure 4). Occasional nuclear pseudoinclusions are present and myxoid or hyalinized stroma may be seen. Mitotic rates of epithelioid schwannoma range from 0–9 mitoses/10 high-power fields (HPFs) in most active areas (mean, 2–3 mitoses/10 HPFs) [62]. Immunohistochemically, the tumor cells are positive for S100 protein. The presence of an EMA-positive perineurial capsule is helpful for diagnosis. Loss of SMARCB1 (IN1) expression is seen in approximately 40% of epithelioid schwannomas [63].

The differential diagnoses of epithelioid schwannoma include myoepithelioma, ossifying fibromyxoid tumor (OFMT), and melanocytic tumors. Myoepitheliomas show reticular, trabecular, or nested growth pattern in myxoid stroma and are positive for epithelial markers. OFMTs are composed of monomorphic ovoid tumor cells arranged in cord or trabecular patterns, with characteristically focal peripheral metaplastic bone formation in myxocollagenous matrix [64]. Melanocytic tumors are unencapsulated and are positive for S100 protein and melanocytic markers.

### 5.2. Granular Cell Tumor

Granular cell tumors are benign neuroectodermal tumors composed of large round to oval cells with abundant and distinctly granular cytoplasm [65]. The tumor can arise at any age, but most cases occur in the fourth to sixth decades of life. Granular cell tumor commonly arises in the extremities, head and neck, and trunk [66]. Loss-of-function mutations in *ATP6AP1* and *ATP6AP2* are present in about 70% of cases [67]. Multiple granular cell tumors have been described in association with Noonan syndrome and LEOPARD syndrome [68,69]. Malignant cutaneous granular cell tumors are rare and usually present as a slowly growing, painless nodule or mass in skin [70].

Histologically, the tumor is composed of sheets, nests, or trabeculae of large, polygonal cells with abundant eosinophilic granular cytoplasm. Tumor cell borders are indistinct. Pseuodoepitheliomatous hyperplasia of the overlying skin is observed in about 30% of cases. Malignant granular cell tumors show nuclear pleomorphism, prominent nucleoli, spindling of cells, frequent mitoses, necrosis, and have high nuclear cytoplasmic ratios [71]. Immunohistochemically, the tumor cells are diffusely positive for S100 protein. Granular cell tumors in subcutaneous tissue express higher levels of TFE3 than those in other sites [72].

The differential diagnoses of granular cell tumor include primitive non-neural granular cell tumor, adult rhabdomyoma, and malignant melanoma. Primitive non-neural granular cell tumors have a polypoid configuration with an epithelial collarette and are negative for S100 protein [73]. Adult rhabdomyomas show sheets of polygonal cells and are positive for desmin and myogenin. Malignant melanomas occasionally show granular cell changes and diffuse positivity for S100 protein, but usually have a junctional component and are positive for melanocytic markers.

### 5.3. Ectopic Meningioma

Ectopic meningioma is a meningothelial neoplasm that occurs entirely outside the anatomical regions that normally contain meningothelial cells, such as intracranial and intraspinal compartments [74]. Cutaneous meningiomas are rare tumors and occur in both congenital and acquired forms [75]. These neoplasms affect all ages but show a slight female predominance. The most common locations are the head and neck region (posterior scalp and sinonasal tract). Cutaneous meningiomas present as a painless, slowly growing mass and are divided into three types based on their clinical and histologic features: Type I-primary cutaneous meningioma, Type II-ectopic meningioma of the soft tissue with extension into skin, and Type III-central nervous system meningioma with extension into skin [76].

Histologically, ectopic meningiomas show epithelioid meningothelial cells arranged in whorls or nests (Figure 5), and structures resembling psammoma bodies may be seen. The histologic appearance of ectopic meningioma is similar to that of primary meningioma of the central nervous system. Examples of all WHO grades (I-III) have been reported, though benign tumors are most common [74]. Immunohistochemically, the tumor cells are positive for EMA and vimentin.

The differential diagnoses of ectopic meningioma include intracranial meningioma, soft tissue perineurioma, cellular neurothekeoma, and plexiform fibrohistiocytic tumor. Extension of intracranial meningioma can be excluded by imaging. Soft tissue perineuriomas lack a nested, lobulated pattern and are positive for CD34. Cellular neurothekeomas show a micronodular growth pattern and are negative for EMA. Plexiform fibrohistiocytic tumors show a mixture of spindled fibroblastic cells and histiocytoid nodules and a plexiform growth pattern [77].

### 5.4. Epithelioid Malignant Peripheral Nerve Sheath Tumor

Epithelioid malignant peripheral nerve sheath tumors (EMPNSTs) are rare variant of MPNST composed of predominantly epithelioid tumor cells (>50%) [78]. Most EMPNSTs are superficial (dermal or subcutaneous) [79,80]. EMPNST differs from conventional MPNST by diffusely expressing S100 protein, having an infrequent association with neurofibromatosis type 1, and having an occasional origin in a schwannoma. The tumor commonly arises in the lower or upper extremities, trunk, or neck. Approximately 80% of EMPNSTs show recurrent *SMARCB1 (INI1)* inactivation [81]. The majority of patients present with a slowly growing, painful, or painless mass.

Histologically, the tumor shows a multilobulated growth pattern. Epithelioid tumor cells are arranged in nests or cords in fibromyxoid stroma. Other histologic features include a rhabdoid cytomorphology, prominent multinucleated giant cells, nuclear inclusions, clear cell morphology, perivascular whorling, and heterologous differentiation [79]. Immunohistochemically, the tumor cells diffusely express S100 protein and SOX10. In contrast to conventional MPNST, the expression of H3K27me3 is intact in EMPNST [82,83]. Loss of SMACR1 (INI1) expression is present in approximately 50% of cutaneous EMPNSTs [84].

The differential diagnoses of EMPNST include epithelioid schwannoma, clear cell sarcoma, malignant melanoma, and carcinoma. Epithelioid schwannomas have smaller tumor cells than EMPNST and lack cytologic atypia, atypical mitoses, and necrosis. Clear cell sarcomas are positive for melanocytic markers and show *EWSR1* gene rearrangement. Malignant melanomas show more nuclear atypia and pleomorphism than EMPNST and are positive for melanocytic markers. Carcinomas are positive for epithelial markers.

## 6. Epithelioid Mesenchymal Tumors of Uncertain Differentiation

### 6.1. Distinctive Dermal Clear Cell Mesenchymal Neoplasm

Distinctive dermal clear cell mesenchymal neoplasm (DDCCMN) is a very rare dermal-based tumor [85]. Its precise line of differentiation has not been determined, and it appears to be benign. DDCCMN usually occurs in adults, and in the majority of reported cases it occurs in lower extremities [86]. Clinically, it presents as a smooth cutaneous nodule.

Histologically, the tumor shows dermal-based nodular proliferation of large polygonal cells with abundant clear cytoplasm and vesicular nuclei (Figure 6). The tumor cells are arranged in sheets and may be markedly atypical with hyperchromatic nuclei [87]. Mitotic activity is rare. Immunohistochemically, the tumor cells are positive for NKI-C3 (CD63) and CD68, but negative for melanocytic and epithelial markers [85].

The differential diagnoses of DDCCMN include perivascular epithelioid cell tumor (PEComa), clear cell adnexal tumors, balloon cell (or clear cell) melanoma, and metastatic clear cell renal cell carcinoma [87]. PEComa shows perivascular epithelioid cells with clear or pale eosinophilic or granular cytoplasm and is positive for smooth muscle and melanocytic markers. Clear cell adnexal tumors show focal adnexal differentiation and are positive for cytokeratin. Balloon cell (or clear cell) melanomas show intraepidermal pagetoid spread and are positive for S100 protein and melanocytic markers [88]. Metastatic clear cell renal cell carcinomas show nests of tumor cells surrounded by prominent delicate vessels and are positive for cytokeratin, EMA, PAX-8, and CA9.

### 6.2. Cellular Neurothekeoma

Cellular neurothekeoma is a distinctive, benign cutaneous neoplasm of uncertain histogenesis [89]. It affects all age groups, but its incidence peaks are in the second and third decades with a female predominance. Cellular neurothekeoma usually arises in an upper limb, head, neck, shoulder, or girdle [90,91]. Clinically, it presents as a solitary, superficial, asymptomatic, slow-growing mass.

Histologically, cellular neurothekeoma is a micronodular or lobulated dermal neoplasm composed of epithelioid to spindle-shaped cells. The tumor cells are arranged in nests separated by dense fibrous septa and have vesicular nuclei and moderate amounts of pale eosinophilic cytoplasm (Figure 7). Myxoid stroma, multinucleated tumor cells, and perineural invasion (neurotropism) are present in some cases. Significant cytologic atypia showing marked nuclear atypia with pleomorphism may be seen [92]. Immunohistochemically, the tumor cells are typically positive for NKI/C3 (CD63), CD10, and PGP9.5. In addition, focal positivity is sometimes noted for SMA and CD68 [91].

The differential diagnoses of cellular neurothekeoma include dermal nerve sheath myxoma, cellular dermatofibroma, plexiform fibrohistiocytic tumor, and spitzoid melanocytic lesions. Dermal nerve sheath myxoma shows hypocellular myxoid lobules with spindle or stellate cells and is positive for S100 protein. Cellular dermatofibroma does not show nested growth patterns. Spitzoid melanocytic lesions show S100 protein and melanoma markers immunoreactivity.

### 6.3. Angiomatoid Fibrous Histiocytoma

Angiomatoid fibrous histiocytoma (AFH) is a distinctive, rarely metastasizing mesenchymal neoplasm of uncertain differentiation [93]. It usually arises in the subcutaneous tissue or deep dermis of the extremities of children and young adults. Most (>90%) of AFHs exhibit t(2;22)(q33;q12), resulting in *EWSR1-CREB1* fusion gene. Clinically, it presents as a painless superficial soft tissue lump, sometimes resembling a hematoma.

Histologically, AFH shows spindle, epithelioid, or histiocytoid cells with nodular and sheet growth pattern. The tumor cells have a moderate amount of palely eosinophilic cytoplasm and ill-defined cell borders. Thick fibrous pseudocapsule and pericapsular cuffing of lymphoplasmacytic cells are present. The blood-filled pseudovascular spaces are surrounded by tumor cells. AFHs may show prominent myxoid stroma, round cell morphology, or pleomorphic cell morphology [93,94,95,96]. Immunohistochemically, the tumor cells are positive for desmin, EMA, and CD99 in about 40%–60% of cases [97]. ALK expression is present in the majority of AFH cases [98].

The differential diagnoses of AFH include aneurysmal fibrous histiocytoma, spindle cell hemangioma, rhabdomyosarcoma, and extraskeletal Ewing sarcoma. Aneurysmal fibrous histiocytomas show overlying epidermal hyperplasia and peripheral entrapment of collagen bundles and are negative for desmin. Spindle cell hemangiomas have true vascular spaces lined by attenuated endothelial cells and are positive for vascular markers. Rhabdomyosarcomas show rhabdomyoblasts or alveolar architecture and are positive for myogenin and MyoD1. Extraskeletal Ewing sarcomas show monotonous small round cells and lack pseudocapsule and pericapsular lymphoplasmacytic cuffing and are positive for NKX2-2 [99].

### 6.4. Myoepithelioma

Cutaneous myoepithelioma is an uncommon neoplasm composed of myoepithelial cells similar to its salivary gland and soft tissue counterparts [100]. The tumor cells are arranged in reticular, trabecular, nested, or sheet-like patterns. Cutaneous syncytial myoepithelioma is a distinct variant showing solid, syncytial growth of histiocytoid to short spindle cells with pale eosinophilic cytoplasm [101]. *EWSR1-PBX3* fusion gene is present in most cases of cutaneous syncytial myoepitheliomas [102], but *EWSR1* rearrangement with unknown fusion partner genes is present in a subset of cutaneous myoepitheliomas [103]. Cutaneous myoepithelioma occurs over a wide age range with male predominance [104]. It commonly arises in the head and neck, and extremities. Clinically, it presents as a painless, slow-growing papule or nodule.

Histologically, the tumor is well-circumscribed, and occasionally has an epidermal collarette. Tumors are lobulated and composed of epithelioid, spindle, plasmacytoid, or clear cells, usually with a myxoid or hyalinized stroma. Moderate to severe nuclear atypia, nuclear pleomorphism, and prominent nucleoli are the most reliable characteristics of myoepithelial carcinoma of soft tissue (Figure 8) [105]. Immunohistochemically, the tumor cells are positive for epithelial markers, S100 protein, and GFAP. In contrast to conventional myoepithelioma of soft tissue, cytokeratin staining is usually negative in cutaneous syncytial myoepithelioma [101].

The differential diagnoses of myoepithelial neoplasms can be broad, depending on predominant histologic patterns. The differential diagnoses of cutaneous myoepithelial neoplasms include EFH, OFMT, extraskeletal myxoid chondrosarcoma, and epithelioid sarcoma. EFHs have a polypod lesion, often with binucleate epithelioid cells, and are positive for ALK. OFMTs show a peripheral rim of metaplastic bone. Extraskeletal myxoid chondrosarcomas have a multinodular architecture and cords, clusters, or a cribriform pattern of tumor cells [106]. Epithelioid sarcomas show epithelioid to spindle cells with eosinophilic cytoplasm and an infiltrative nodular growth pattern.

### 6.5. Epithelioid Sarcoma

Epithelioid sarcoma is a rare soft tissue sarcoma composed of epithelioid cells with an epithelial phenotype [107] and has two subtypes: the classic (distal) and proximal type. The classic type often involves skin and commonly occurs in a distal upper extremity (particularly in fingers or hands), distal lower extremity, or head and neck [108,109]. The proximal type tends to arise in deep soft tissue and often affects the pelvis, perineal, and pubic regions [110]. Both types affect patients of various ages. In both types, the *SMARCB1 (INI1)* gene located at 22q11 is inactivated, which correlates with loss of SMARCB1 (INI1) expression by immunohistochemistry [111,112]. Clinically, the classic type usually presents as one or more indurated, ill-defined, dermal, or subcutaneous nodules or masses.

Histologically, epithelioid sarcomas are composed of atypical epithelioid and plump spindle-shaped cells arranged in vague nodules (Figure 9). Central necrosis and myxohyaline changes may be seen. The tumor cells have abundant eosinophilic cytoplasm and vesicular nuclei. Proximal-type epithelioid sarcoma is characterized predominantly by large cells with epithelioid morphology and marked cytologic atypia with frequent rhabdoid features [110]. Immunohistochemically, the tumor cells of proximal-type epithelioid sarcoma are positive for cytokeratin and EMA. CD34 is expressed in approximately 60% of cases [113].

The differential diagnoses of epithelioid sarcoma are wide, and include granulomatous lesions, pseudomyogenic hemangioendothelioma, epithelioid angiosarcoma, extrarenal rhabdoid tumor, malignant melanoma, and carcinomas. Granulomatous lesions lack cytologic atypia and cytokeratin expression. Pseudomyogenic hemangioendothelioma shows myogenic morphology with plump spindle cells and bright eosinophilic cytoplasm and does not express CD34. Epithelioid angiosarcomas show at least a focal vasoformative growth pattern and are positive for CD31 and ERG. Extrarenal rhabdoid tumors usually occur in infants and young children and have a predominant rhabdoid morphology. Malignant melanomas are positive for S100 protein, SOX10, and melanocytic markers. Primary or metastatic carcinomas are positive for cytokeratin and EMA and retain SMARCB1 (INI1) expression.

### 6.6. Dermal Clear Cell Sarcoma

Dermal clear cell sarcoma is a malignant neoplasm of uncertain histogenesis that exhibits melanocytic differentiation [114]. It occurs predominantly in young adults [115], and males and females are equally affected [116]. Dermal clear cell sarcoma commonly arises in the foot, lower extremities, or hip. Recurrent *EWSR1-ATF1* gene fusion from t(12;22)(q13;q12) is present, and an alternative *EWSR1-CREB1* gene fusion from t(2;22)(q33;q12) has also been identified in a small subset of cases [117,118]. Clinically, dermal clear cell sarcoma presents as a slowly, enlarging mass.

Histologically, the tumor is composed of fascicles and nests of uniform spindled to epithelioid cells, with intervening collagen. The cytoplasm is clear to pale eosinophilic, and multinucleated wreath-like giant cells are often present. Melanin pigment is seen in two-thirds of cases. Compound variants of clear cell sarcoma with intraepidermal involvement have been described [119,120]. Immunohistochemically, the tumor cells are positive for S100 protein and melanocytic markers.

The differential diagnoses of dermal clear cell sarcoma include malignant melanoma, EMPNST, and PEComa. It is extremely difficult to distinguish malignant melanoma from dermal clear cell sarcoma, though malignant melanoma shows a junctional component and greater cytologic atypia. *BRAF* mutations are observed in about 52% of malignant melanoma cases [121], but *EWSR1* rearrangement has not been identified in malignant melanoma. Thus, molecular analyses can exclude malignant melanoma. EMPNST does not contain melanin and is negative for melanocytic markers. PEComa co-expresses smooth muscle and melanocytic markers.

### 6.7. PEComa

PEComas are mesenchymal neoplasms composed of distinctive cells that associate focally with blood vessel walls and usually express melanocytic and smooth muscle markers [122]. PEComas show a wide anatomical distribution, but most often arise in the retroperitoneum, uterus, gastrointestinal tract, or abdominopelvic region. Primary cutaneous PEComas are relatively rare, have a wide age range, and arise predominantly in lower extremities [123,124,125]. Clinically, these lesions usually present as a single painless, slow-growing nodule or plaque.

Histologically, the tumor is composed of epithelioid cells with abundant granular eosinophilic or clear cytoplasm and round to oval nuclei (Figure 10). The tumor cells are arranged in nested, trabecular, or sheet patterns. Thin-walled capillary vessels are present between nests. Sclerosing PEComa and fibroma-like PEComa variants have been described [126,127]. Malignant PEComas are characterized by large tumor size (>5 cm), infiltrative growth, a high-nuclear grade, necrosis, and mitosis rates (>1/50 HPFs) [128]. Immunohistochemically, the tumor cells are positive for melanocytic markers (with HMB-45 being the most sensitive marker) and smooth muscle markers (e.g., SMA, calponin). In contrast to visceral PEComas, cutaneous PEComas do not harbor *TFE* gene fusions and consistently lack TFE3 expression [129]. Metastatic clear cell renal cell carcinomas show nests of tumor cells surrounded by prominent delicate sinusoidal vessels and are positive for cytokeratin, EMA, PAX-8, and CA9.

The differential diagnoses of cutaneous PEComa includes clear cell dermatofibroma, melanocytic tumor, and metastatic clear cell renal cell carcinoma. Clear cell dermatofibromas display a storiform arrangement of spindle cells and do not express melanocytic markers [130]. Melanocytic tumors, such as balloon cell nevi and balloon cell (or clear cell) melanomas, usually show a junctional component and express melanocytic markers.

## 7. Other Tumors with Epithelioid Features

Atypical fibroxanthoma (AFX), pleomorphic dermal sarcoma (PDS), and dermatofibrosarcoma protuberans (DFSP) may show epithelioid morphologies. AFX is a dermally based neoplasm of uncertain histogenesis [131]. Histologically, AFX shows a nodular or polypoid tumor composed of atypical pleomorphic, spindle, and epithelioid cells with overlying epidermal collarette. Histologic variants of AFX include spindle cell, clear cell, granular cell, keloidal, myxoid, sclerotic, pseudoangiomatous, and variants with osteoclast-giant cell, osteoid, or chondroid formation [132]. AFX is a diagnosis of exclusion; markers of specific differentiation must be negative [131]. PDS is an undifferentiated pleomorphic tumor [133]. Histologically, PDS shows pleomorphic, epithelioid, spindle, and multinucleated giant cells. AFX and PDS are genetically related, potentially representing two ends of a common tumor spectrum [134]. PDS can be distinguished from AFX by subcutaneous invasion, tumor necrosis, and perineural or lymphovascular invasion [135,136]. DFSP is a superficial, low grade, locally aggressive neoplasm characterized by monomorphic spindle cells arranged in storiform pattern and honeycomb pattern of subcutaneous infiltration [137]. DFSP can show broad morphological variation, including myxoid change, granular cell change, myoid differentiation, pigmented dendritic cells, giant cell fibroblastoma-like morphology, and fibrosarcomatous transformation [138,139]. Recognition of variable morphological patterns in DFSP is important to avoid misdiagnosis.

Epithelioid variants have been described in angiosarcoma [140], myxofibrosarcoma [141], pleomorphic liposarcoma [142], and dedifferentiated liposarcoma [143]. Cutaneous non-mesenchymal neoplasms, especially, including malignant melanomas, adnexal tumors, sarcomatoid carcinomas, and metastatic carcinomas, may mimic epithelioid mesenchymal neoplasms. Differentiation can be made using a combination of clinical findings, diagnostic histologic features, appropriate immunohistochemical panels, and molecular analyses.

## 8. Conclusions

Herein we provide a practical diagnostic approach to epithelioid cutaneous mesenchymal neoplasms and review their histologic features. Epithelioid cutaneous mesenchymal neoplasms exhibit broad and divergent lineages and considerable overlapping histologic features. Malignant melanoma and primary and metastatic carcinoma should be considered in differential diagnoses, and careful histologic evaluations, immunohistochemical analyses, and molecular diagnostic studies can aid correct diagnoses. Strict histologic criteria should always be used for proper diagnoses. Recent advances in the identification of genetic alterations and immunohistochemical markers will improve the accuracy of diagnosis and avoid misdiagnosis.

## Figures and Tables

**Figure 1 diagnostics-10-00233-f001:**
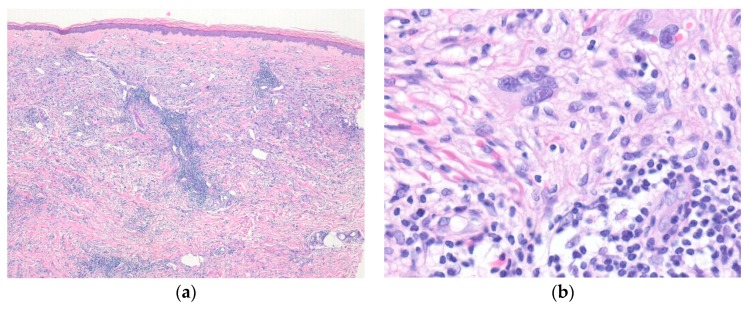
Myxoinflammatory fibroblastic sarcoma. (**a**) A poorly defined lesion in dermis shows prominent inflammatory cell infiltrates and scattered mono- or multinucleated pleomorphic cells; (**b**) Occasional pleomorphic multinucleated giant cells with viral inclusion-like nucleoli are seen in the background of inflammatory cell infiltrates (H&E stain, original magnifications 40× (**a**) and 200× (**b**)).

**Figure 2 diagnostics-10-00233-f002:**
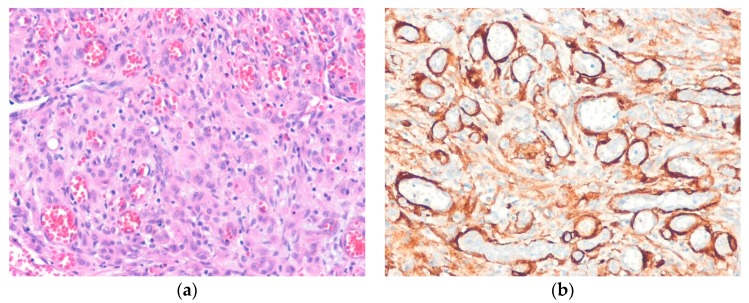
Cellular epithelioid hemangioma. (**a**) The tumor shows epithelioid endothelial cells with a solid pattern; (**b**) Immunostaining for smooth muscle actin highlights pericytes and well-formed blood vessels (H&E stain, original magnification 200× (**a**); SMA immunostain, original magnification 200× (**b**)).

**Figure 3 diagnostics-10-00233-f003:**
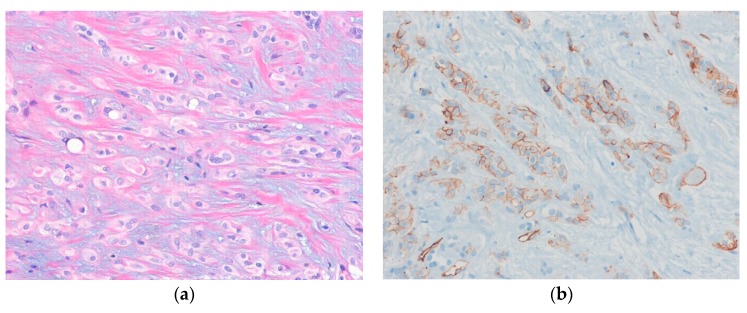
Epithelioid hemangioendothelioma. (**a**) The tumor cells of epithelioid hemangioendothelioma are arranged in trabecular or cord-like patterns in a myxohyaline stroma. Occasional intracytoplasmic vacuoles are present; (**b**) The tumor cells are positive for CD31 (H&E stain, original magnification 200× (**a**); CD31 immunostain, original magnification 200× (**b**)).

**Figure 4 diagnostics-10-00233-f004:**
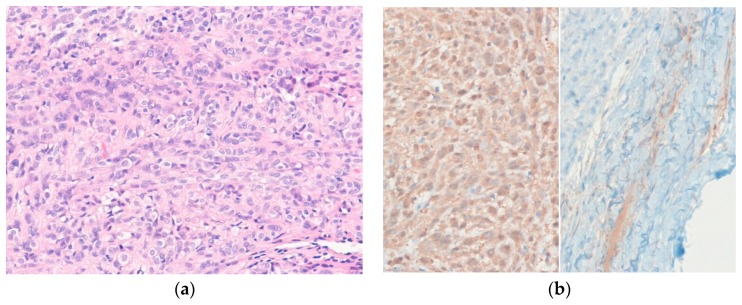
Epithelioid schwannoma. (**a**) The bland epithelioid tumor cells are arranged in sheets and nests; (**b**) The tumor cells are diffusely positive for S100 protein (left). EMA highlights the perineurial capsule (right) (H&E stain, original magnification 200× (**a**); S100 protein and EMA immunostain, original magnification 200× (**b**)).

**Figure 5 diagnostics-10-00233-f005:**
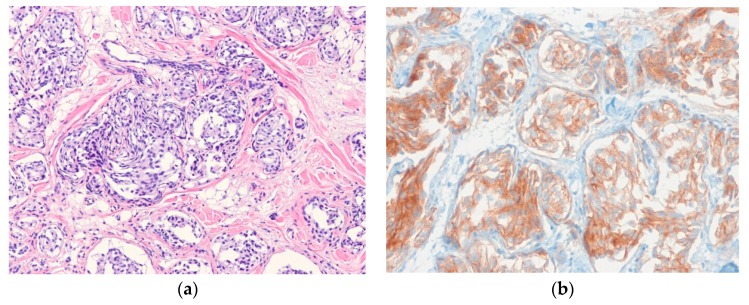
Ectopic meningioma. (**a**) The tumor cells are arranged in whorling or nesting pattern, with fibromyxoid stroma; (**b**) The tumor cells are positive for EMA (H&E stain, original magnification 100× (**a**); EMA immunostain, original magnification 200× (**b**)).

**Figure 6 diagnostics-10-00233-f006:**
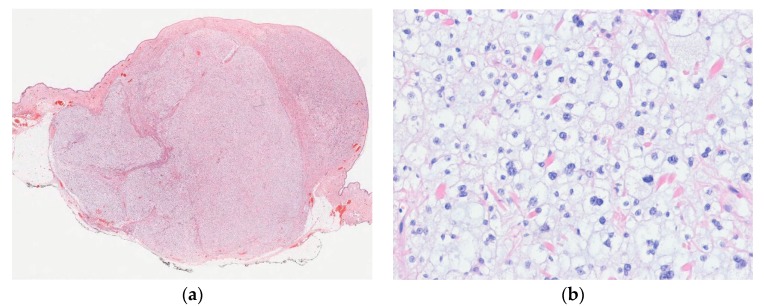
Distinctive dermal clear cell mesenchymal neoplasm. (**a**) Scanning magnification shows a well-circumscribed, solitary tumor in the dermis and subcutaneous tissue; (**b**) The tumor cells have epithelioid or polygonal cells with abundant clear cytoplasm and are arranged in sheet-like pattern (H&E stain, original magnifications 10× (**a**) and 200× (**b**)).

**Figure 7 diagnostics-10-00233-f007:**
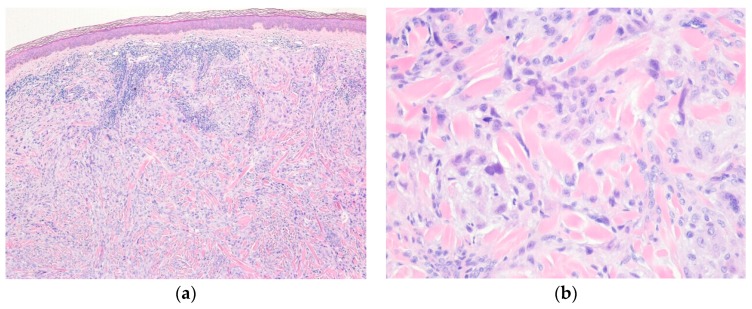
Cellular neurothekeoma. (**a**) At lower power, the tumor shows a nested growth pattern separated by sclerotic collagen; (**b**) The epithelioid tumor cells have pale eosinophilic cytoplasm. Mild nuclear atypia is also present (H&E stain, original magnifications 40× (**a**) and 200× (**b**)).

**Figure 8 diagnostics-10-00233-f008:**
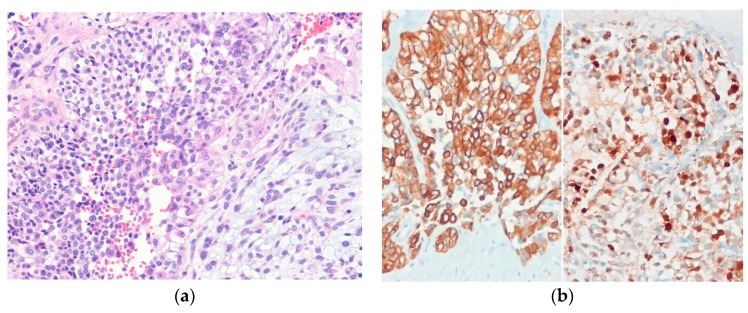
Myoepithelial carcinoma. (**a**) The tumor cells have cohesive epithelioid appearance with nuclear atypia and prominent nucleoli; (**b**) The tumor cells are positive for cytokeratin (AE1/AE3) (left) and S100 protein (right) (H&E stain. original magnification 100× (**a**); cytokeratin (AE1/AE3) and S100 protein immunostain, original magnification 200× (**b**)).

**Figure 9 diagnostics-10-00233-f009:**
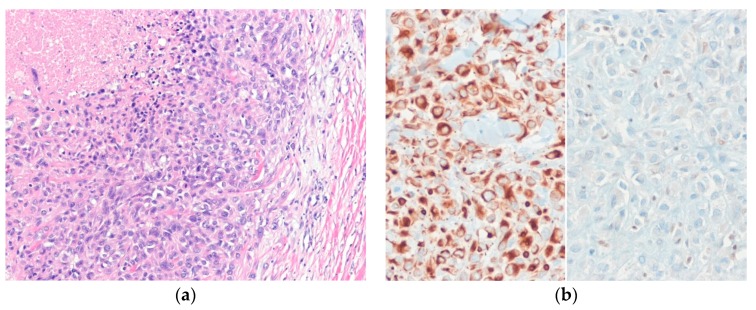
Classic-type epithelioid sarcoma. (**a**) The tumor shows sheets of epithelioid cells with eosinophilic cytoplasm and vesicular nuclei and central necrosis; (**b**) The tumor cells are positive for cytokeratin (AE1/AE3) (left) and show loss of SMARCB1 (INI1) expression (right) (H&E stain, original magnification 100× (**a**); cytokeratin (AE1/AE3) and SMARCB1 (INI1) immunostain, original magnification 200× (**b**)).

**Figure 10 diagnostics-10-00233-f010:**
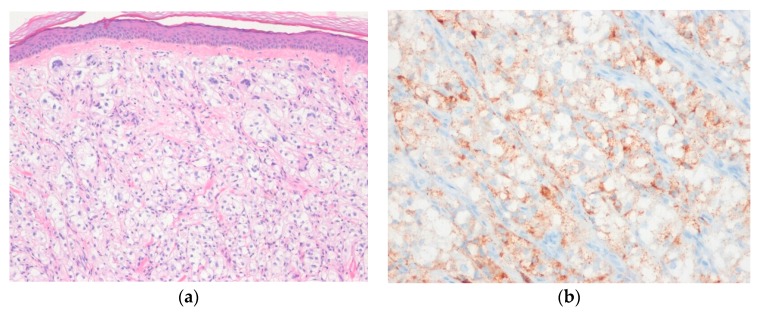
Cutaneous PEComa. (**a**) The tumor shows a nested architecture with epithelioid tumor cells containing abundant clear or granular eosinophilic cytoplasm; (**b**) The tumor cells are positive for HMB45 (H&E stain, original magnification 100× (**a**); HMB-45 immunostain, original magnification 200× (**b**)).

**Table 1 diagnostics-10-00233-t001:** Architectural patterns of epithelioid cutaneous mesenchymal neoplasms.

**Lobulated Architecture**	
	Epithelioid hemangioma
	Myoepithelioma
	Epithelioid MPNST
**Nested Architecture**	
	Cellular neurothekeoma
	Ectopic meningioma
	Dermal clear cell sarcoma
	PEComa
**Trabecular and Cord-Like Architecture**	
	Myoepithelioma
	Epithelioid schwannoma
	Epithelioid hemangioendothelioma
	Sclerosing PEComa
**Sheet-Like Architecture**	
	Epithelioid fibrous histiocytoma
	Cutaneous leiomyosarcoma (epithelioid variant)
	Glomus tumor
	Epithelioid schwannoma
	Granular cell tumor
	Distinctive dermal clear cell mesenchymal neoplasm
	Angiomatoid fibrous histiocytoma
	Epithelioid sarcoma
**Clear or Granular Cells**	
	Granular cell tumor
	Distinctive dermal clear cell mesenchymal neoplasm
	Myoepithelioma
	Dermal clear cell sarcoma
	PEComa
**Prominent Inflammatory Cells**	
	Myxoinflammatory fibroblastic sarcoma
	Pseudomyogenic hemangioendothelioma (subset)
	Epithelioid hemangioma

MPNST, malignant peripheral nerve sheath tumor; PEComa, perivascular epithelioid tumor.

**Table 2 diagnostics-10-00233-t002:** Immunoreactivity of selected epithelioid cutaneous neoplasms.

	Epithelioid Fibrous Histiocytoma	Pseudomyogenic Hemangio-endothelioma	Myoepithelioma/Myoepithelial Carcinoma	Epithelioid MPNST	Epithelioid Sarcoma	PEComa	Malignant Melanoma	Carcinoma
CD34	−	−	−	+/−	+/−	−	−	−
SMA	−	−	+/−	−	+/−	+	−	−
S100 protein	−	−	+	+		+	+	−
ALK	+	−	−	−	−	−	−	−
SMARCB1 (INI1)	+	+	+/−	+/−	−	−	+	+
Vascular markers (e.g., CD31, ERG)	−	+	−	−	−	−	−	−
Melanocytic markers (e.g., HMB45, Melan-A)	−	−	−	−	−	+	+	−
Epithelial markers (e.g., Cytokeratin (AE1/AE3), EMA)	−	+	+	+/−	+	−	−	+

MPNST, malignant peripheral nerve sheath tumor; PEComa, perivascular epithelioid tumor; +, positive staining; +/−, variable staining; −, negative staining.

**Table 3 diagnostics-10-00233-t003:** Recently identified cytogenetic and molecular alterations in cutaneous epithelioid mesenchymal neoplasms.

Neoplasm	Cytogenetic Alterations	Molecular Alterations
Epithelioid fibrous histiocytoma	2p23	*VCL-ALK* *SQSTM1-ALK*
Myxoinflammatory fibroblastic sarcoma	t(1;10)(p22;q24) 3p11-12 amplification	*TGFBR3-MGEA5* *VGLL3, CHMP2B* amplification
Glomus tumor		*MIR143-NOTCH* *NOTCH* rearrangement
Epithelioid hemangioma		*ZFP36-FOSB* *WWTR1-FOSB*
Pseudomyogenic hemangioendothelioma	t(7;19)(q22;q13)	*SERPINE* *1-FOSB*
Epithelioid hemangioendothelioma	t(1;3)(p36;q25) t(X;11)(p11;q22)	*WWTR1-CAMTA1* *YAP1-TFE3*
Granular cell tumor		*ATP6AP1, ATP6AP2* mutation
Epithelioid MPNST (subset)		*SMARCB1 (INI1)* mutation
Angiomatoid fibrous histiocytoma	t(2;22)(q33;q12) t(12;22)(q13;q12) t(12;16)(q13;p11)	*EWSR1-CREB1* *EWSR1-ATF1* *FUS-ATF1*
Cutaneous syncytial myoepithelioma		*EWSR1-PBX3*
Epithelioid sarcoma		*SMARCB1 (INI1)* deletion
Dermal clear cell sarcoma	t(12;22)(q13;q12) t(2;22)(q33;q12)	*EWSR1-AFT1* *EWSR1-CREB1*

MPNST, malignant peripheral nerve sheath tumor.

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
