# Peer review of "Epithelioid Cutaneous Mesenchymal Neoplasms: A Practical Diagnostic Approach"

_diagnostics, 2020, doi:10.3390/diagnostics10040233_

Round 1

Reviewer 1 Report

Manuscript entitled "Epithelioid Cutaneous Mesenchymal Neoplasms: A Practical Diagnostic Approach". This work is informative and comprehensive. I would suggest the following modifications before final acceptance:

  1. Angiomatoid fibrous histiocytoma should be included.
  2. Certain cutaneous myogenic tumors should be included.

Author Response

Reviewer 1's comment: 

  I would suggest the following modifications before final acceptance:

  1. Angiomatoid fibrous histiocytoma should be included.
  2. Certain cutaneous myogenic tumors should be included.

Author's response:

 We have included angiomatoid fibrous histiocytoma and cutaneous myogenic   tumors (smooth muscle tumors).

Reviewer 2 Report

Complete and well-written review. The paper does not contain elements of novelty, but it is an excellent review of histological morphology and immunohistochemistry of uncommon mesenchymal neoplasms. I can suggest to add some paragraphs about other entities, maybe more common and more of interest for the readers of the Journal, such as atypical fibroxanthoma, pleomorphic dermal sarcoma and dermato-fibrosarcoma protuberans.

Author Response

Reviewer 2' s comment:

 I can suggest to add some paragraphs about other entities, maybe more common  and more of interest for the readers of the Journal, such as atypical  fibroxanthoma, pleomorphic dermal sarcoma and dermatofibrosarcoma protuberans.

Author's response:

  We have add some paragraphs describing atypical fibroxanthoma, pleomorphic dermal sarcoma, and dermatofibrosarcoma protuberans.